# Student Burnout in Higher Education: From Lockdowns to Classrooms

**Kevin Michael Jackson [1]** and **Márta Konczosné Szombathelyi [2,*]**

1 Business and Administration Sciences Department, Széchenyi István University, 9026 Gyor, Hungary
2 Leadership and Marketing Department, Faculty of Economics, Széchenyi István University, 9026 Gyor, Hungary
* Correspondence: kszm@sze.hu

**Abstract:** During the spring 2021 semester, COVID-19 forced most universities around the world to teach exclusively online in a very short time frame. This situation reversed itself, however, during the fall 2021 semester when COVID-19 restrictions were lifted as teachers and students returned to classrooms. This study includes ninety-seven international students who participated in surveys at the beginning and the end of the fall 2021 semester, which included questions related to burnout, self-efficacy, resiliency, home environments, and technical issues. Students were asked to reflect on their educational experiences during the spring 2021 and fall 2021 semesters. The purpose of this study is to identify and examine the most significant changes that occurred between these two semesters. The results indicate a significant shift in student burnout as challenges with home environments were replaced with ones related to returning to the classroom. Even as the concerns about COVID-19 lessen, higher education institutions must understand the magnitude and permanence of its impact.

**Keywords:** higher education; COVID-19; students' burnout; online learning; face-to-face learning; hybrid learning

## 1. Introduction

Herbert Freudenberger was a well-known psychologist and among the first to coin the phrase "burnout" and conduct comprehensive studies to better understand its root causes. He characterized burnout as the tendency "to fail, wear out, or become exhausted by making excessive demands on energy, strength, or resources" [1]. According to Freudenberger, the symptoms of burnout vary from one person to the next and can be controlled, but not eliminated. Christina Maslach and Susan Jackson made a significant contribution to the study of burnout in 1981 with the creation of the Maslach Burnout Inventory model (MBI). The MBI model is a tool that helps researchers assess the burnout of individuals who are suffering from emotional exhaustion and cynicism that occurs from engaging in "people work" [2]. Since its inception, this model has evolved significantly and the introduction of the MBI–General Survey [3] extended its relevance to include additional professions and occupational groups [4]. While students are not actively employed, their psychological situation due to academic requirements can be considered "work" [5]. The MBI model has been widely used in higher education for decades and COVID-19 has unfortunately further strengthened its relevance. Numerous research studies have also shown that the MBI model has produced data on the burnout and engagement of university students that were both reliable and reinforced the validity of the three-tier construct [6].

Student burnout had been a steady topic of research well before the COVID-19 pandemic. Jacobs and Dodd [7] conducted a study on 149 American university students using MBI to assess student burnout as a function of personality, social support, and workload. Surveys were given at the beginning of the fall Semester and the beginning of the following spring semester to measure the change in student sentiment during this period. Their

findings concluded that personalities, such as negative temperament, can increase the likelihood of burnout in some students. Although academic workload can lead to emotional exhaustion, this study also showed that the definition of workload is often subjective and not an effective measure of burnout. In a similar study, Daniel Law [8] similarly surveyed 163 American business students to test the reliability of the MBI model's three components: emotional exhaustion, depersonalization, and personal accomplishment. The findings from this study indicate that students experienced higher levels of exhaustion and burnout at the end of the semester, mainly due to exams, and that there is a risk that burnout can carry over to the next semester [9–11]. This phenomenon was later supported by a Spanish study that found burnout levels were elevated by COVID-19 lockdowns and remained elevated for long periods of time [12]. In a large MBI study completed in Brazil and Portugal [13], the results showed that burnout was best characterized by factors that relate to physical and psychological exhaustion and ones that address cynicism and disengagement. These results are consistent with the findings of other burnout studies [14,15], where fatigue and exhaustion were found to be core reasons for burnout. It was interesting that self-efficacy was not found to be a significant third factor for burnout in this study. A larger MBI study was conducted on 7757 Italian university students [16]. In contrast to the previous study, the three-factor structure was confirmed and was invariant for male and females and undergraduates and graduates. Exhaustion was found to be a significant cause of burnout that can be mitigated by managing academic demands. A meta-analysis was completed in the UK in 2020 that included over 100,000 students from twenty-nine different studies. Its purpose was to examine the relationship between burnout and academic achievement. The results of this meta-analysis revealed that the burnout symptoms of exhaustion, cynicism, and reduced efficacy were all significantly negatively correlated with academic achievement [17].

Many research studies have concluded that physical and psychological exhaustion are leading causes of student burnout. The COVID-19 pandemic had the unfortunate effect of amplifying the burnout effects of university students by having a significant and negative impact on mental health, education, and the daily routines of students [18,19]. A UK study investigated the impact of COVID-19 on the mental health of 214 university students [20]. The overall study lasted from October 14th, 2019, to January 28th, 2020, and included two surveys before the UK lockdown and two surveys after the UK lockdown. The results show that mental well-being and physical activity decreased during the lockdown, while perceived stress and sedentary time increased. A COVID-19 study completed in Turkey surveyed 485 students for the purpose of examining the direct and indirect relationships between student stress and burnout, depression, and well-being [21]. The results of this study also indicate that COVID-19 did significantly contribute to student burnout, depression, and negative effects on their psychological well-being. A recent and similar study of 199 Polish university students showed that academic burnout during COVID-19 had an indirect effect on post-traumatic stress disorder (PTSD) that was mediated by significant levels of anxiety and fear [22]. Finally, a recent study in Finland conducted three research studies from May 2020 to May 2021 to examine how the burnout and engagement of university students (1501, 1526, 1685) changed during this period. The results show that there was significant volatility from one semester to the next as burnout peaked in April 2021, whereas student engagement reached a low point in December 2020 [23]. This important research demonstrates how educational institutions operating in an uncertain world need to collect data more often to be able to properly address the needs of their students.

Academic self-efficacy refers to a student's belief in their own capabilities and their ability to achieve a certain level of academic performance [24,25]. This means that students who display a higher level of self-efficacy for completing an educational task are more likely to participate more, work harder, and better endure hardships than those who question their own capabilities [26]. While there have been skeptics of how to measure self-efficacy, it has proven to be pervasive across a multitude of research studies and conceptually and empirically distinct from a wide range of related constructs [27]. In 1995,

Schwarzer and Jerusalem [28] created the General Self-Efficacy Scale (GSE) that consists of items designed for the general adult population. The validity of the GSE model has been consistently affirmed by studies since its inception. A large-scale study was completed on 19,120 participants across 25 countries and once again reaffirmed the construct [29]. Another significant study was conducted among 1,933 myocardial and tumor removal patients in Germany, Poland, and South Korea [30] who were experiencing significant amounts of stress and anxiety. Once again, the GSE model proved to be a universal construct that relates well to other constructs. A Turkish study of 354 university students discovered that the relationship between GSE and life satisfaction and burnout was significant [31]. Another Turkish study of 120 university students found a strong negative correlation between self-efficacy and burnout [32].

The COVID-19 pandemic has had an enormous impact on all levels of education worldwide. A recent study conducted on 756 nursing students in Poland, Spain, and Slovakia revealed that while a high level of generalized self-efficacy was observed, there were significant differences between resident countries [33]. Two large studies conducted at American universities surveyed students regarding the impact of emergency remote instruction (EMI) on self-efficacy [34]. The overall results revealed that students showed a 50% loss of efficacy beliefs after EMI but were able to improve to 75% following instructor interventions. Finally, a literature review of various journals, books, and government publications found that the levels of self-efficacy can vary significantly, and that a student's level of self-efficacy is heavily influenced by the level of parental, teacher, classmate, and close friend support [35].

The purpose of this paper is to measure, analyze, and compare the burnout of higher education students in the spring 2021 and fall 2021 semesters. Ninety-seven students from twenty-six different nationalities participated in both the BOS and EOS surveys where Germany and France were the most represented. The courses Consumer Behavior (undergraduate), Services Marketing (undergraduate), Services Marketing (Master's), and Entrepreneurship (undergraduate) were taught at Corvinus University and Digital Management (undergraduate) was taught at the ESSCA School of Management during the fall 2021 semester. Approximately 40% of the students were master's students and 60% were undergraduate students. Of the 97 students in the sample, 34 were male and 67 were female. The mean age of the students was 23 years old, the youngest student was 18, and the oldest student was 27 years old. 82.5% of the surveyed students were between twenty and twenty-four years old.

This paper will address the following hypotheses:

1.  The COVID-19 lockdowns during the spring 2021 semester negatively and disproportionately affected the burnout of higher education students with lower remote learning and home environment sentiments.
2.  The return to the classroom during the fall 2021 semester will positively impact the burnout of students who struggled with remote learning.
3.  The level of a student's self-efficacy remains significantly and negatively correlated to burnout during the spring 2021 and fall 2021 semesters.

The rest of the paper will be organized as follows: Section 2 outlines the Materials and Methods, Section 3 presents and discusses the Results, Section 4 provides the Discussion of the results, and Section 5 offers the Conclusion, and possible future research options.

## 2. Materials and Methods

The following theoretical framework served as the basis for carrying out the study now presented to understand how the root causes of student burnout evolved during the fall 2020, spring 2021, and fall 2021 semesters.

### 2.1. Spring 2021 Semester Survey Design

At the beginning of the spring 2021 semester, it was clear that the spread of COVID-19 was far from over and that lockdowns would once again be mandated in most universities

around the world. Due to these extraordinary circumstances, we initiated a research project to better understand the impact that this pandemic would have on higher education students during the fall 2020 and spring 2021 semesters. Due to the unprecedented and rapidly changing conditions, surveys were given at the beginning of the spring 2021 semester (students reflected back on the fall 2020 semester) and at the end of the spring 2021 semester (students reflected back on the spring 2021 semester). Students in the aforementioned courses were presented with the research topic in class, given the opportunity to ask questions, and asked for their voluntary participation. The students who chose to participate in these surveys did so by clicking a link to surveyplanet.com that was posted in the relevant Microsoft Teams course groups. It should be noted that 83 students voluntarily participated in both the BOS and EOS surveys. The spring 2021 surveys are shown in Supplementary Materials (SM1).

### 2.2. Fall 2021 Semester Study Design

The fall 2021 semester surveys were created using experience gained from the spring 2021 surveys. Students were similarly asked to participate in the fall BOS survey (students reflected back on the spring 2021 semester) and EOS survey (students reflected back on the fall 2021 semester) on a voluntary basis using the Surveyplanet platform. Ninety-seven students voluntarily participated in both the BOS and EOS surveys. The survey sections burnout (BUR), resiliency (RES), self-efficacy (GSE), technical situation (OED), COVID-19 (COV), remote learning sentiment (HME), and hybrid learning sentiment (HYB) were added and all used a Likert scale (1–5). The fall 2021 semester surveys can be found in SM2 and the abbreviations and terminology can be found in SM3.

### 2.3. Instrument

To create a survey that was both comprehensive and expedient, the following choices were made in the respective survey sections.

### 2.3.1. Maslach Burnout Inventory (BUR)

During the lockdowns of the spring 2021 semester, it was clear from our experiences and experiences shared by others that many students were experiencing burnout symptoms. While the Maslach Burnout Inventory (MBI) was originally intended to measure the burnout of workers, many studies have demonstrated its relevance to higher education students where "study" may also be considered "work." A key consideration in selecting specific points from the MBI Inventory was the emphasis on the return to the classroom during the fall 2021 semester and the reactions of students to this transition. There were also overlapping questions from the HME section and the MBI Inventory meaning that fewer items were selected. Items from the depersonalization dimension were excluded as their primary focus is on employment. Please see SM4 for the selected and adapted points from the MBI Burnout Inventory and SM5 for the BOS/EOS data reliability statistics.

### 2.3.2. Connor–Davidson Resilience Scale (RES)

Based on the experience gained from the spring 2021 surveys, the transition back to the classroom, and the questions from the other survey sections, ten items from the Connor–Davidson Resilience Scale [36] were selected. The aim was to maintain a balance between comprehensiveness and expediency. Please refer to SM6 for the selected items and SM7 for the BOS/EOS data reliability statistics.

### 2.3.3. General Self-Efficacy Scale (GSE)

All the ten GSE statements were included in the surveys. Please see SM8 for these GSE statements and SM9 for the BOS/EOS data reliability statistics.

### 2.3.4. Online Education (OED)

Based on the experience gained from the spring 2021 semester, these items were selected to understand how student perception of remote learning changed during the fall 2021 semester while students returned to the classroom. Please see SM10 for the BOS/EOS data reliability statistics.

### 2.3.5. COVID-19 Response (COV)

This section was designed to measure how the students felt about their home universities' response to COVID-19 and their views on COVID-19 at the start of the fall 2021 semester. Please see SM11 for the BOS/EOS data reliability statistics.

### 2.3.6. Home Environment (HME) (USM)

This section was also used in the spring 2021 surveys and is designed to measure the impact of home learning on students and its disparity with traditional in-class learning. In the fall 2021 EOS survey, however, students had returned to the classroom and were receiving university services directly. University self-management (USM) reflects this significant shift from remote learning to the return to the classroom. Please see SM12 for the BOS/EOS data reliability statistics.

### 2.3.7. Hybrid Learning (HYB)

The transition from the spring 2021 semester to the fall 2021 semester was historic as it went from learning exclusively online to the return to the classroom. This section was designed to measure the changes in student perception toward hybrid learning given their exposure to the extremes of lockdowns and then their transition back to the classroom. Please see SM13 for the BOS data reliability statistics.

### 2.4. Data Reliability

Please refer to Table 1 for a summary of the data reliability of independent variables from the BOS and EOS surveys.

**Table 1.** Data reliability summary of independent variables (BOS, EOS).

| | Beginning of the Semester (BOS) | | End of the Semester (EOS) | |
|---|---|---|---|---|
| | **Excluded Questions** | **Overall Cronbach's $\alpha$** | **Excluded Questions** | **Overall Cronbach's $\alpha$** |
| Burnout (BUR) | 2, 5 | 0.677 | 2, 5 | 0.732 |
| Resiliency (RES) | - | 0.784 | - | 0.81 |
| General Efficacy (GSE) | - | 0.823 | - | 0.799 |
| Online Education (OED) | 7 | 0.676 | 7 | 0.659 |
| COVID–19 Response (COV) | 2, 3 | 0.698 | 2, 3 | 0.69 |
| Home Environment (HME) (USM) | 2, 3, 6, 8, 9, 10, 11, 12 | 0.69 | 2, 8, 9, 11 | 0.903 |
| Hybrid Learning (HYB) | 1, 2, 3, 5, 6, 8, 9, 12 | 0.892 | - | - |

Note: Of the observations, pairwise complete cases were used. The following items were reverse scaled: BOS_HME4, BOS_HME7, and BOS_HME13. For the EOS (HME) and BOS (HYB) sets of questions, principal factor analysis (PCA) was used. Source: own data.

## 3. Results

### 3.1. BOS and EOS Burnout Correlations

The effect sizes in psychological research are often misinterpreted and underappreciated. To interpret the following data, we will assume that an effect size $r = 0.05$ is very small, $r = 0.10$ is small but more consequential, $r = 0.20$ indicates a medium effect offering some explanation, $r = 0.30$ indicates a large effect potentially powerful in the short and long term, and $r = 0.40$ or greater is potentially an overestimate [37].

As a first step, Spearman's correlation coefficients were calculated between burnout and other independent variables from the BOS and EOS surveys. Please note that Home Environment in the fall 2021 BOS survey was changed to University Self-Management in the fall EOS survey since the place of learning shifted from homes to classrooms and the direct impact of university services became significantly greater.

### 3.2. Results and Hypotheses

#### 3.2.1. Hypothesis 1: Student Burnout, and Remote Learning Sentiment

The first hypothesis states: "The COVID-19 lockdowns during the spring 2021 semester negatively and disproportionately affected the burnout of higher education students with lower remote learning and home environment sentiments." Referring to Table 2, one can see that the remote learning sentiment of university students in the fall 2021 (BOS) was negatively and moderately correlated with burnout ($r = -0.209$, $p = 0.035$).

**Table 2.** Correlations of independent variables with burnout using Spearman's method.

| Variables | Fall 2021 BOS Burnout | | Fall 2021 EOS Burnout | |
|---|---|---|---|---|
| | r | *p*-Value | r | *p*-Value |
| Age | −0.136 | 0.173 | −0.192 | 0.062 |
| Self-Efficacy | −0.320 | $p = 0.001$ | −0.128 | $p = 0.211$ |
| Resiliency | −0.392 | $p \leq 0.001$ | −0.128 | $p = 0.051$ |
| Home Environment + | −0.288 | $p = 0.003$ | - | - |
| University Self-Management | - | - | −0.302 | $p = 0.003$ |
| Hybrid Preference | −0.206 | $p = 0.038$ | 0.123 | $p = 0.23$ |
| Remote Learning Sentiment | −0.209 | $p = 0.035$ | 0.424 | $p \leq 0.001$ |
| Online Learning Preference | −0.178 | $p = 0.074$ | 0.399 | $p \leq 0.001$ |
| Technical Support | −0.157 | 0.114 | −0.365 | $p \leq 0.001$ |

Note: BOS to BOS and EOS to EOS correlations are used. Source: own data.

During the COVID-19 lockdown, it is reasonable to conclude that students with more negative views of remote learning were also more likely to experience burnout symptoms. As a significant amount of research suggests, there are a wide variety of factors that can influence university student burnout and COVID-19 has added even more complexity to this equation. The data from the fall BOS survey also revealed that home environment sentiment showed an even stronger correlation ($r = -0.288$, $p \leq 0.001$) with burnout. In the fall BOS, there was a strong, positive correlation observed between Home Environment and Remote Learning Sentiment ($r = 0.581$, $p \leq 0.001$).

#### 3.2.2. Hypothesis 2: Student Burnout, and the Return to the Classroom

The second hypothesis states: "The return to the classroom during the fall 2021 semester will positively impact the burnout of students who struggled with remote learning." Strong, positive correlations were observed in the fall 2021 EOS survey between remote learning sentiment and burnout ($r = 0.424$, $p \leq 0.001$) and online learning preference and burnout ($r = 0.399$, $p \leq 0.001$). These positive correlations stand in contrast to the negative correlations observed in the fall 2021 BOS. On one side, there are the students who experienced burnout symptoms while returning to the classroom. Potential causes of this burnout are the cost and time of commuting, loss of flexibility, less comfort, less interaction, and increased social pressure. On the other side, there are the students who were happy to be back in the classroom and this is evidenced by the negative correlations between university self-management and burnout ($r = -0.302$, $p = 0.003$) and technical support and burnout ($r = -0.365$, $p \leq 0.001$) (Table 3). This suggests that better support services offered by universities lowered the burnout rates for many students. While a higher remote sentiment helped shield students from burnout in the fall 2021 BOS, it became a source of burnout for

these same students in the fall 2021 EOS survey. Overall, the data suggest that there are distinct groups of students who have very different views regarding their education.

**Table 3.** Correlations of independent variables with technical support using Spearman's method.

| Variables | Fall 2021 BOS Technical | | Fall 2021 EOS Technical | |
|---|---|---|---|---|
| | r | *p*-Value | r | *p*-Value |
| Age | 0.045 | $p = 0.65$ | 0.121 | 0.241 |
| Home Environment + | 0.327 | $p \leq 0.001$ | - | - |
| University Self-Management | - | - | 0.439 | $p \leq 0.001$ |
| Hybrid Learning Preference | 0.219 | $p = 0.027$ | 0.078 | $p = 0.446$ |
| Remote Learning Sentiment | 0.347 | $p \leq 0.001$ | −0.399 | $p \leq 0.001$ |
| Online Learning Preference | 0.289 | $p = 0.003$ | −0.293 | $p \leq 0.004$ |

Note: BOS to BOS and EOS to EOS correlations are used. Source: own data.

### 3.2.3. Hypothesis 3: Student Burnout, and Self-Efficacy

The third hypothesis states: "The level of a student's self-efficacy remains significantly and negatively correlated to burnout during the spring 2021 and fall 2021 semesters." Self-efficacy was significantly and negatively correlated ($r = −0.320$, $p = 0.001$) to burnout in the fall 2021 BOS. This result was expected as research studies have consistently shown that higher self-efficacy is a burnout deterrent. In the EOS survey, however, self-efficacy remained negative correlated ($r = −0.128$, $p = 0.211$), but weaker and less significant. For the students who were happy to return to the classroom, their self-efficacy became less important as university support services were easier to access thereby reducing stress and anxiety. The students who were unhappy with the return to the classroom and experienced burnout symptoms and this had the effect of lowering their perceived self-efficacy. While Hypothesis 3 holds true for the fall 2021 BOS, the return to the classroom had the effect of reducing the negative correlation between self-efficacy and burnout due to either the substation of university services or the loss of self-efficacy from students who were unhappy with the transition back to the classroom.

### 3.3. Factors Affecting Home Environments (BOS)

In Table 4, some correlation can be seen between age and fall 2021 BOS home environment ($r = 0.359$, $p \leq 0.001$). One explanation for this correlation is that master's students are older on average and on two-year programs and the foreign, undergraduate students are only attending for one semester. The transition in and out of another school in a foreign country during one semester often involves challenges. We can also see that the students who have a high remote learning sentiment and prefer online learning were more likely to be happier with their home environments during the COVID-19 lockdown and less likely to experience burnout symptoms. This result lends further support to Hypothesis 1.

**Table 4.** Correlations of independent variables with home environment using Spearman's method.

| Variables | BOS Home Environment + | |
|---|---|---|
| | r | *p*-Value |
| Age | 0.359 | $p \leq 0.001$ |
| Remote Learning Sentiment | 0.581 | $p \leq 0.001$ |
| Online Learning Preference | 0.566 | $p \leq 0.001$ |
| Hybrid Learning Preference | 0.244 | $p = 0.014$ |

Source: own data.

### 3.4. The Importance of University Self-Management (EOS)

In Table 5, a strong, negative correlation can be observed between remote learning sentiment and university self-management (USM) ($r = −0513$, $p \leq 0.001$) and online learning preference and university self-management ($r = −0.658$, $p \leq 0.001$). This tells us that the

students who have high levels of remote learning sentiment and highly prefer online learning did not value the services and support offered by the university during the return to the classroom. This result lends further support to Hypothesis 2.

**Table 5.** Correlations of independent variables with university self-management (USM) using Spearman's method.

| Variables | EOS University Self-Management | |
| --- | --- | --- |
| | r | *p*-Value |
| Age | 0.021 | $p = 0.842$ |
| Remote Learning Sentiment | −0.513 | $p \leq 0.001$ |
| Online Learning Preference | −0.658 | $p \leq 0.001$ |
| Hybrid Learning Preference | −0.046 | $p = 0.654$ |

Source: own data.

### 3.5. BOS Burnout Linear Regression

The first model (M1) is a multiple linear regression analysis designed to predict burnout based on age and gender. A regression equation was found ($F_{(2, 99)} = 1.14$, $p < 0.324$), with an $R^2$ of 0.023 explaining 2.3% of the sample variance. This analysis indicates that age and gender are not significant predictors of student burnout based on BOS survey data. In the second model (M2), self-efficacy was added to age and gender creating the regression equation ($F_{(3, 98)} = 3.92$, $p < 0.011$), with an $R^2$ of 0.107 explaining 10.7% of the variance. One can see that the addition of self-efficacy created a more significant model with a higher explained variance. For the third model (M3), remote learning sentiment was added creating the regression equation ($F_{(4, 97)} = 4.25$, $p < 0.003$), with an $R^2$ of 0.149 explaining 14.9% of the variance. The effect of self-efficacy remained constant, and it is evident that remote learning sentiment did also have a positive impact on the explained variance between M2 and M3. In fourth model (M4), home environment sentiment was added creating the regression equation ($F_{(5, 96)} = 4.188$, $p < 0.002$), with an $R^2$ of 0.179 explaining 17.9% of the variance. The effect of self-efficacy remained constant between M3 and M4 lending support to Hypothesis 3. The addition of home environment sentiment, however, did have an impact on the remote learning sentiment between M3 and M4. This suggests that there is significant overlap between remote learning sentiment and home environment sentiment (Table 6). Please refer to SM14 for the complete BOS and EOS regression models.

**Table 6.** Explained variance and association between the selected independent variables and burnout (BOS).

| Variables | BOS M1 ($R^2 = 0.023$) | BOS M2 ($R^2 = 0.107$) | BOS M3 ($R^2 = 0.149$) | BOS M4 ($R^2 = 0.179$) |
| --- | --- | --- | --- | --- |
| Age | −0.047 [ns] | 0.034 [ns] | 0.043 [ns] | 0.109 [ns] |
| Gender (Female) | 0.234 [ns] | 0.177 [ns] | 0.149 [ns] | 0.146 [ns] |
| Self-Efficacy | | −0.303 ** | −0.303 ** | −0.293 ** |
| Remote Sentiment | | | −0.206 * | −0.085 [ns] |
| Home Sentiment | | | | −0.224 [T] |

[ns], [T] $p < 0.1$; * $p < 0.05$; ** $p < 0.01$. Source: own data.

### 3.6. EOS Burnout Linear Regression

In the first model (M1), the regression equation ($F_{(2, 92)} = 1.23$, $p < 0.296$) with an $R^2$ of 0.026 explains 2.6% of the sample variance. This analysis indicates that age and gender are not significant predictors of student burnout using EOS survey data. In the second model (M2), self-efficacy was added to age and gender creating the regression equation ($F_{(3, 91)} = 1.173$, $p < 0.324$), with an $R^2$ of 0.037 explaining 3.7% of the variance.

Self-efficacy was far less significant in the EOS survey than in the BOS. For the third model (M3), university self-management was added creating the regression equation ($F_{(4, 90)} = 4.348$, $p < 0.003$), with an $R^2$ of 0.162 explaining 16.2% of the variance. The addition of this university self-management significantly boosted the explained variance, while self-efficacy did not have a significant impact on M3. This starkly contrasts with the influence of self-efficacy in the BOS regression model. In fourth model (M4), technical sentiment was added creating the regression equation ($F_{(5, 89)} = 5.012$, $p < 0.001$), with an $R^2$ of 0.22 explaining 22% of the variance. The addition of technical sentiment was significant, and it had a negative effect on university self-management. Self-efficacy was again not significant in M4, which does not support Hypothesis 3. The fifth model (M5) added remote learning sentiment creating the regression equation ($F_{(6, 88)} = 6.064$, $p < 0.001$), with an $R^2$ of 0.174 explaining 29.3% of the variance. The addition of remote learning significantly and negatively affected the significance of university self-management where technical support was affected but to a lesser extent (Table 7)

**Table 7.** Explained variance and association between the selected independent variables and burnout (EOS).

| | EOS M1 ($R^2 = 0.026$) | EOS M2 ($R^2 = 0.037$) | EOS M3 ($R^2 = 0.162$) | EOS M4 ($R^2 = 0.22$) | EOS M5 ($R^2 = 0.293$) |
|---|---|---|---|---|---|
| Age | −0.156 [ns] | −0.109 [ns] | −0.091 [ns] | −0.117 [ns] | −0.146 [ns] |
| Gender (Female) | 0.091 [ns] | 0.056 [ns] | 0.056 [ns] | 0.046 [ns] | 0.044 [ns] |
| Self-Efficacy | | −0.117 [ns] | −0.085 [ns] | 0.011 [ns] | −0.013 [ns] |
| Uni Self Manage | | | −0.356 [***] | −0.211 [*] | −0.054 [ns] |
| Technical Sentiment | | | | −0.299 [**] | −0.228 [*] |
| Remote Sentiment | | | | | 0.336 [**] |

[ns] $p < 0.1$; [*] $p < 0.05$; [**] $p < 0.01$; [***] $p < 0.001$. Source: own data.

### 3.7. BOS Burnout Mediation

In the BOS mediation analysis, self-efficacy and remote learning sentiment are independent variables, home environment sentiment is the mediating variable, and burnout is the dependent variable (Figure 1). The results show that the direct effects of remote learning sentiment on burnout produced a slight negative correlation ($\beta = -0.076$ [ns]). The indirect effect between remote learning sentiment and burnout mediated by home environment sentiment ($\beta = -0.09$ [**]) again shows a slight negative correlation. The total effect of remote learning sentiment on burnout is ($\beta = -0.166$ [**]) indicating a moderate, negative correlation. These data tell us that a higher remote sentiment can moderately lessen burnout effects thus lending support to Hypothesis 1. Please refer to SM15 for the complete BOS and EOS mediation models.

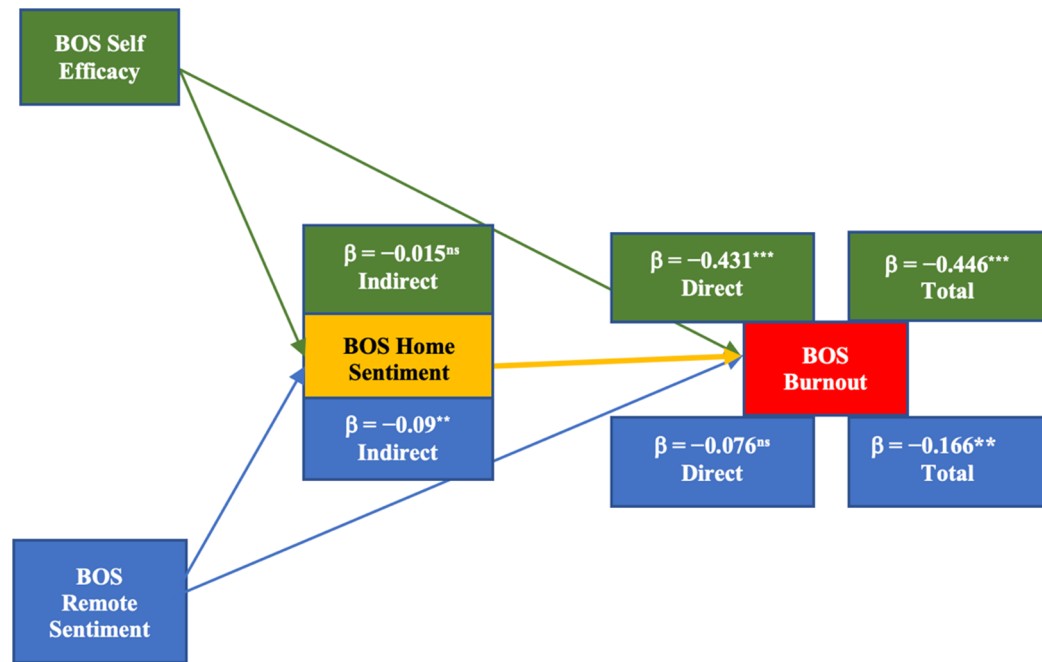

**Figure 1.** BOS flowchart of direct and indirect burnout effects using beta values. Note: $^{ns}$ $p < 0.1$; ** $p < 0.01$; *** $p < 0.001$. Source: own data.

### 3.8. EOS Burnout Mediation

In the EOS mediation analysis, university self-management is the independent variable, remote learning sentiment is the mediating variable, and burnout is the dependent variable (Table 8 and Figure 2). The results show the direct effects of university self-management on burnout produced a moderate, negative correlation ( $\beta = -0.174$ **). The indirect effect between university self-management mediated by remote learning sentiment shows a moderate, negative correlation ( $\beta = -0.151$ **). The total effect of university self-management on burnout is ( $\beta = -0.325$ ***) indicating a moderate, negative correlation. These data show that both the direct and indirect effects of university self-management on burnout are moderately, negatively correlated. The total effects, however, do indicate that university self-management is fairly broad as it has direct burnout effects and indirect burnout effects mediated by remote learning sentiment.

**Table 8.** The direct and indirect burnout effects between burnout and selected independent variables.

| | Direct Burnout Effects | Indirect Burnout Effects | Total Effects |
|---|---|---|---|
| BOS Remote Sentiment | $\beta = -0.076$ $^{ns}$ | - | - |
| BOS Self-Efficacy | $\beta\beta = -0.431$ *** | - | - |
| BOS Remote Sentiment Home Environment Mediation | - | $\beta = -0.09$ ** | $\beta = -0.166$ ** |
| BOS Self-Efficacy Home Environment Mediation | - | $\beta = -0.015^{ns}$ ** | $\beta\beta = -0.446$ *** |
| EOS University Self-Management | $\beta\beta = -0.174$ ** | - | - |
| EOS University Self-Management Remote Sentiment Mediation | - | $\beta\beta = -0.151$ *** | $\beta\beta = -0.325$ *** |

$^{ns}$ $p < 0.1$; ** $p < 0.01$; *** $p < 0.001$. Source: own data.

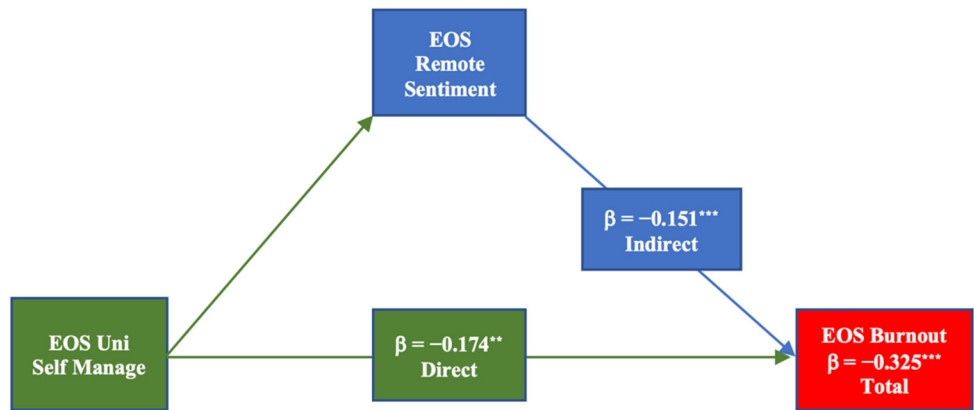

**Figure 2.** EOS direct and indirect burnout effects. Note: ** $p < 0.01$; *** $p < 0.001$. Source: own data.

## 4. Discussion

### 4.1. A Tale of Two Semesters

The rapid transition from face-to-face learning to online learning during the COVID-19 pandemic was unprecedented. While many students briefly returned to the classroom during the fall 2020 semester, this return was short-lived as a spike in global COVID-19 cases [38], particularly in Europe and North America, forced universities to teach exclusively online during the spring 2021 semester. Many research studies over the past year have provided compelling evidence regarding the correlation between COVID-19 and student burnout. This correlation, however, is very complex as burnout disproportionately affects specific groups of students. One recent study in Germany [39] reported that while most students experienced low to moderate burnout symptoms since the start of the COVID-19 pandemic, some students experienced severe symptoms that required urgent attention. Another recent study measured the burnout of medical students and residents in Belgium during the first COVID-19 lockdown [40] and found that those who perceived a higher impact of COVID-19 also experienced higher levels of burnout on all dimensions related to their studies. These studies lend support to the fact that student burnout cannot be resolved using a "one size fits all approach."

While student burnout during the COVID-19 lockdowns is well documented, the burnout associated with the transition back to the classroom is not. The research presented in this paper shows that students with higher remote learning sentiments were more likely to experience burnout symptoms as they returned to the classroom. Students with lower remote learning senitments embraced the transition back to the classroom and consequently experienced less or reduced burnout symptoms. Remote learning sentiment, therefore, is one important variable that can help universities better understand the evolving sources of burnout and how to address them. Self-efficacy was a more relevant variable to burnout during the spring 2021 semester, but became less of a factor during the fall 2021 semester as many students took more comfort in the services provided by their universities. Similarly, resiliency proved to more significantly correlated to burnout during the lockdowns of the spring 2021 semester, but became less correlated following the return to the classroom during the fall 2021 semester. The data from this research study indicate that variables affecting student burnout changed significantly between a short time frame of two semesters and that there are distinct groups of students when identifying sources of burnout.

### 4.2. Recommendations

#### 4.2.1. Measuring and Addressing Burnout

Since the topic of burnout is quite complex, researchers, Mäkikangas and Kinnunen [41] developed a five-step approach towards creating personal burnout profiles for working people. A similar approach was created Leiter and Maslach [42] calling for a person-centered approach to properly assess burnout in the workplace. Given the complexity of student

burnout and the uncertainty present in a post COVID-19 world, universities should create and maintain student burnout profiles in a similar fashion. Such burnout profiles would prove to be very useful if conditions change dramatically similar to what occurred during the spring 2021 and fall 2021 semesters.

### 4.2.2. Hybrid Learning as the Great Compromise

As the research from this study shows, while COVID-19 has forced most university students to learn online, the range of remote learning sentiments varies significantly. Some students learn better in-class and some students prefer to learn online. Hybrid or blended learning is being increasingly seen by higher education as a compromise that satisfies the needs of the greatest number of students. The data in Table 9 show that a significant number of students in the fall 2021 semester felt that hybrid learning best described their mode of education. Many studies conducted during COVID-19 have concluded that hybrid or blended learning represents a way to be prepared for future disruptions and to better prepare students for a world that will inevitably use more technology [43–46].

**Table 9.** Mode of education in spring 2021 and fall 2021 semesters (n = 97).

|  | Spring 2021 | Fall 2021 |
| --- | --- | --- |
| Entirely Online | 66 | 1 |
| Hybrid Learning | 24 | 49 |
| Entirely in the Classroom | 4 | 45 |
| Internships | 3 | 2 |
| Total | 97 | 97 |

Source: own data.

### 4.2.3. The "New Normal" Is Permanent

The term "new normal" is now used exhaustively and one of the reasons why is because it is still extremely difficult to define. The transition from face-face learning to online learning was very significant for global higher education. The transition from online learning to face-face learning is also very significant and it is clear that higher education will not be able to return to its 2019 state. It is certain that uncertainty will characterize the "new normal" and this makes understanding student burnout more difficult to idenitfy and manage. Higher education institutions should create student burnout profiles and update them regularly. Hybrid learning has the potential to alleviate the burnout of many students by creating a balance between face-to-face and online learning similar to the growing trend of working from home [47,48].

### 5. Conclusions

While COVID-19 has created unprecedented challenges for higher education, many higher education institutions view the digitization of education as more of an opportunity and less of a burden [49,50]. Technology will enable higher education institutions to gather data about students more frequently and comprehensively so they can proactively mitigate sources of burnout. These burnout profiles are another step in the direction of personalized learning, which is consistent with student experiences outside of the classroom.

Given what we have learned from the COVID-19 pandemic, learning exclusively online has serious drawbacks for a significant percentage of students. Returning to how things were in 2019, however, it also not practical or optimal as an enormous amount of change has taken place during the past two years. Hybrid learning on a discipline-by-discipline basis can be an effective way for higher education institutions to accommodate the needs of diverse groups of students, recognize the lasting role of technology in education, and remain consistent with permanent work from home trends in a post COVID-19 world.

Despite the challenges that higher education has endured since the beginning of the COVID-19 pandemic, there is good reason to believe that it will act as a catalyst for better, more accessible, and more affordable education.

## 6. Study Limitations and Future Work

All the students participating in the surveys were taught by one professor, Kevin Jackson. In the future, it will be preferable to conduct research using numerous instructors along with a diverse group of students. A larger sample size would yield better results and a more compelling analysis. Items from the Maslach Burnout Inventory (MBI) and the Connor–Davidson Resilience Scale were selected, and others were omitted. These choices could have had an impact on the viability of the results.

The COVID-19 pandemic has created a "new normal" that is characterized by a significant increase in uncertainty for higher education institutions and their students. Future work will include conducting studies on the effects of creating student burnout profiles and their impact on mitigating the effects of burnout. It will also be interesting in the future to investigate the viability of the adaptability scale [51] and the expected negative correlation between higher adaptability scores and lower burnout rates.

**Supplementary Materials:** The following supporting information can be downloaded at: https://www.mdpi.com/article/10.3390/educsci12120842/s1, Supplementary Material 1 (SM1): Spring 2021 Beginning of the Semester (BOS) and End of the Semester (EOS) Surveys; Supplementary Material 2 (SM2): Fall 2021 Beginning of the Semester (BOS) and End of the Semester (EOS) Surveys; Supplementary Material 3 (SM3): Abbreviations/Terminology; Supplementary Material 4 (SM4): Items from the Maslach Burnout Inventory; Supplementary Material 5 (SM5): BOS and EOS Burnout Unidimensional Reliability (BUR); Supplementary Material 6 (SM6): Connor–Davidson Resilience Scale; Supplementary Material 7 (SM7): BOS and EOS Resiliency Unidimensional Reliability (RES); Supplementary Material 8 (SM8): Generalized Self-Efficacy Scale (GSE); Supplementary Material 9 (SM9): BOS and EOS General Self Efficacy Unidimensional Reliability (GSE); Supplementary Material 10 (SM10): BOS and EOS Online Education Unidimensional Reliability (OED); Supplementary Material 11 (SM11): BOS and EOS COVID-19 University Response Unidimensional Reliability (OED); Supplementary Material 12 (SM12): BOS Home Environment and EOS University Self-Management Unidimensional Reliability (HME) (USM); Supplementary Material 13 (SM13): BOS Hybrid Learning Unidimensional Reliability (HYB); Supplementary Material 14 (SM14): Fall 2021 BOS and EOS Burnout Regression Analysis; Supplementary Material 15 (SM15): Fall 2021 BOS and EOS Burnout Mediation Analysis.

**Author Contributions:** Conceptualization, K.M.J.; methodology, K.M.J. and M.K.S.; software, K.M.J.; validation, K.M.J. and M.K.S.; formal analysis, K.M.J.; investigation, K.M.J.; resources, K.M.J.; data curation, K.M.J.; writing—original draft preparation, K.M.J.; writing—review and editing, K.M.J. and M.K.S.; visualization, K.M.J.; supervision, M.K.S.; project administration, K.M.J. All authors have read and agreed to the published version of the manuscript.

**Funding:** This research received no external funding.

**Institutional Review Board Statement:** Ethical review and approval were waived for this study because the results did not involve any personal data, only anonymous aggregated data were used.

**Informed Consent Statement:** Informed consent was obtained from all subjects involved in the study.

**Data Availability Statement:** The data presented in this study are available from the corresponding author on reasonable request.

**Acknowledgments:** We thank the students whose participation made this research possible.

**Conflicts of Interest:** The authors declare no conflict of interest.

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
