# Peer review of "Student Burnout in Higher Education: From Lockdowns to Classrooms"

_education, doi:10.3390/educsci12120842_

Round 1
Reviewer 1 Report
Dear authors,
I believe your work has the potential to contribute to the important matter of burnout among university students. The rationale behind your research is interesting. However, there are several methodological limitations that - I am afraid - undermine the validity of your work. I hope with my remarks to help you to improve the manuscript when possible.
- line 62-64: the Italian study you have mentioned do not make such a statement. The article has only referred to other studies in the introduction. I suggest you more attention while citing articles to make sure they actually support your thesis.
- line 76: burnout is multifactorial, and as you state yourself later on in the manuscript, many factors contribute to its onset. I would suggest changing "cause" into "contribute".
- line 109-110: I recommend you to be consistent in indicating the Country of the university rather than the specific institutions.
- line 121-122: I appreciate your awareness of the limitation in the choice of certain methods; however, they should be all mentioned and discussed properly in the appropriate section of the manuscript (typically at the end of the discussion). The limitations of the article are fundamental to correctly frame the findings of your research.
- line 138: it may appear like the study is multicenter. I would suggest you changing "from twenty-six countries" in " of 26 nationalities".
- line 148-158: this part does not concern the methods of the manuscript. It seems more appropriate to move it in the introduction or in the discussion.
- methods: in this section, it is essential to provide information about recruitment (invitation by email?), data collection (online survey?), inclusion/exclusion criteria (if any). In a second paragraph, it is important to clearly describe the variables, with precise definition, how they were assessed, how the scores were used (e.g., standardization?). In some case you have used labels for variables that were not included in the table 2. University self-management is not described. Is it a self-report measure? How was that assessed? Why was it mentioned for the first time directly under table 3?
- burnout: how was burnout assessed and estimated? The MBI is composed by three domains (it seems you only picked two) with independent scores. How have you combined them? You need to be more transparent in your analysis, and comply to the guidelines of open science.
- questionnaires: A rationale behind the exclusion of items should also be included. Questionnaires are validated with their specific structure. You are not supposed to only pick some of the items. If you do, I expect a stronger rationale than the time limitation. If the time is an issue (I am aware it is), then less variables should be included. Decreasing the quality of the assessment undermine the validity of your study.
- appendix: if you refer to the appendix at the end of the manuscript, I suggest you refer to the supplementary file with a different label. You have two appendix A, B, C and D attached to your work.
- line 117-118: this information should be mentioned in detail in the methods section, while describing the data collection. Then, it should be clear how information was used. You mentioned in the introduction and in table 1 the spring 2021, after that you always mentioned only data from fall 2021. What happened to the spring one? How many time points do you have? Two or four? Beginning and end of the two semesters? It is very confusing. Please, try to clarify this aspect to help the reader.
- statistical analysis: in the methods section you should describe all the statistical analysis you have performed with the underlying rationale. You have mentioned part of it in the results section (e.g., line 231-232, line 233-234), but some is still missing (e.g., how the correlation was estimated? I suppose with Spearman's rho, but unclear). I suggest you a parallel structure between statistical analysis and results. In this way the reader can easily follow each step of your analysis and read the subsequent results.
- tables: each table should be understandable independently from the rest of the manuscript. In the title you should clearly state what the table is, not what coefficients are showed. That should be showed in the table.
- table 2: what is the "overall" Cronbach's alpha?
- table 2: the structure may be improved (see the example in the attached file).
- table 3: I understand you refer to direct correlation to distinguish from the indirect one later on, but then I would frame "correlations of independent variables with burnout estimated with Spearman's method (?)".
- table 7-8: the titles of the tables should explain what the table is about. Example: "Explained variance and association between the selected independent variables and burnout at the beginning of the semester".
The measures, as well as the dependent variable, should be showed in the table (see the example in the attached file)
- statistical significance: I would recommend you consider the inclusion of the 95% CI or some measure of uncertainty in your results rather than "ns".
- Discussion: I read with interest your arguments in the discussion section. However, I would suggest you frame your reasoning within the literature with more references when possible. Moreover, in some case, even if reasonable, your arguments are a bit too far from the evidence of your study. For example, point 4.4.4 (line 384) seems more a general consideration about corona time and consequences on labor market rather than a consequence of the associations you have investigated.
- line 312-313: you have written "students specifically mentioned". I believe you have not conducted open interview and what you refer to is probably coming from the questionnaires (Q16). Then, I would consider providing the information about the frequency of these specific items, something like "x% of students reported that...".
- Gender: You have asked students to inform about their gender. How have you assessed that? Have you included non-binary as an option? More and more guidelines are recommending that. See: Nature journals raise the bar on sex and gender reporting in research
- Appendix F: clarify the meaning of bold characters. Not only in appendix F that is not clear.

Reviewer 2 Report
Thank you for the opportunity to review the paper entitled "Student Burnout in Higher Education: From Lockdowns to Classrooms".
This is a work of interest in that it focuses on relatively little studied psychological aspects of the role that burnout has played in situations of online and hybrid teaching during the occurrence of the covid-19 pandemic.
And although the work presents relevant aspects, it presents deficiencies in its methodological and structural approach. Fortunately, these problems are manageable and therefore the authors could reach a compatible solution, from my point of view, with the feasibility of the work with a view to publication.
The introduction is sufficient to support the objectives of the study. The bibliographic sources consulted are relevant, recent and sufficient.
The main deficiencies are in the Materials and method section.
Authors should insert a procedure-focused subsection within the method section. It should explain in some detail how the investigation was carried out. The information included in section 2 must allow the eventual replication of the study.
The "survey models" section presents some doubts. The authors must indicate the scientifically based reasons on which they are based to select the questions they have used (and not only the saving of time). The alteration of the original instruments could affect the validity and reliability of the evaluation instruments, so evidence of these aspects should be provided, beyond the Overall Cronbach's α
A subsection explaining the statistical analyzes carried out can also be inserted at the end of section 2.
The results section could be structured to respond to the hypotheses raised. One suggestion is to compose this section in subsections focused on the hypotheses. Thus, the structure of this section would be adapted to the discussion section.
References to other studies that have focused on the subject could be included in the discussion section (or more broadly in the conclusions), indicating whether or not some of the results coincide with previous evidence. The authors could expand a little more on the conclusions of the study.
With all this and to conclude, this report aims to help the authors. Being a study of interest and with good and original research approaches, the work can be accepted for publication with slight changes.
I congratulate the authors for the work done and I encourage them to carry out the suggested changes and to continue in this line of work.
Reviewer 3 Report
1. The references need important organization based by journal criteria.
2. I recommended to introduce more recent references (from 2020 to 2022) related to student burnout.
3. I recommended to include more future solutions, resulting from data analysis, to combat the phenomenon of student burnout.
